# The Effect of an Er, Cr: YSGG Laser Combined with Implantoplasty Treatment on Implant Surface Roughness and Morphologic Analysis: A Pilot In Vitro Study

**DOI:** 10.3390/jfb13030133

**Published:** 2022-08-29

**Authors:** Chih-Jen Lin, Ming-Hsu Tsai, Yu-Ling Wu, Hsuan Lung, Hung-Shyong Chen, Aaron Yu-Jen Wu

**Affiliations:** 1Department of Dentistry, Kaohsiung Chang Gung Memorial Hospital and Chang Gung University College of Medicine, Kaohsiung 833, Taiwan; 2Department of Mechanical Engineering, Cheng Shiu University, Kaohsiung 833, Taiwan; 3Center for Environmental Toxin and Emerging-Contaminant Research, Cheng Shiu University, Kaohsiung 833, Taiwan

**Keywords:** Er, Cr: YSGG Laser, implantitis, implantoplasty, TiUnite surface

## Abstract

Although laser irradiation and implantoplasty (IP) are both treatment options for peri-implantitis, no studies have yet combined these two treatment solutions. The aim of this study was to identify the effect of an Er, Cr: YSGG laser on the IP surface. In experiment 1, TiUnite anodized surface implants were treated with an Er, Cr: YSGG laser at 0.5 to 2 W on the panel energy setting and 20 Hz under water irrigation. In experiment 2, all implant surfaces were treated with the IP procedure first, then irradiated with the Er, Cr: YSGG laser. All samples were analyzed by stereomicroscopy, scanning electron microscopy (SEM), energy dispersive X-ray spectroscopy (EDS), and surface topography. Stereomicroscopy and SEM revealed no obvious surface change at any energy setting once the surface was polished with the IP procedure, whereas damage was caused to the TiUnite original implant surface when the Er, Cr: YSGG laser panel energy was set at 1 W or higher. EDS showed no significant difference in element composition once the surface was polished with the IP procedure, while a compositional change was detected when the Er, Cr: YSGG laser panel energy was set to 0.5 W or higher to irradiate the original TiUnite surface. Surface roughness may be related to laser irradiation energy, but no significant changes occurred following IP. These results indicated that the Er, Cr: YSGG laser may have little effect on the post-IP surface compared with the virgin TiUnite surface.

## 1. Introduction

Dental implants had become a popular choice of substitution for missing dentitions. Even though they have a high success rate [1,2], some osseointegrated implants fail inevitably for various reasons, including peri-implantitis [3], overloading due to inappropriate biomechanical force [4], and improper implantation sites. The peri-implant disease can be divided into peri-mucositis and peri-implantitis. While peri-mucositis is the presence of peri-implant signs of inflammation combined with no additional bone loss following initial healing, peri-implantitis is a pathological condition characterized by inflammation in the peri-implant connective tissue and progressive loss of supporting bone [5,6,7]. Peri-implantitis is one of the major risk factors affecting the long-term success of dental implants [8]. Factors that may be related to peri-implantitis include poor oral hygiene and plaque control, history of periodontitis, smoking, diabetes, excess cement, and occlusal overload [7,9].

Nonsurgical treatment should always be the priority before surgical intervention [10], and different mechanical instruments can be used for the elimination of an established biofilm on the implant surface. Mechanical instruments made of stainless metals can easily cause damage to the titanium surface of the implant fixture and abutment [11,12].

Recently, various types of lasers were introduced for implant surface debridement in nonsurgical therapy [13,14]. The Er: YAG laser, used with water irrigation, was able to remove subgingival calculus effectively from the tooth surface [15]. Other results showed that Er: YAG irradiation promoted a significant increase in oxides and a decrease in microscopical roughness and porosity on sandblasted and acid-etched titanium disks [16]. Lasers are also expected to help tissues in an inflamed or damaged state rapidly enter the healing and regenerative phases by thorough debridement and decontamination of diseased tissues and by modulating or activating cell metabolism in the surrounding tissues [17,18]. Some studies have shown that morphological changes can be observed when 30 mJ of Er: YAG laser is applied to an implant without water irrigation [12]. Of the available Erbium laser technologies, the Er, Cr: YSGG laser operates at a wavelength of 2780 nm, which is different from that of the Er: YAG laser (2940 nm), and the range of available pulse durations is also different. The Er, Cr: YSGG laser also has the potential for removing calcified deposits from the implant surface [19,20], and with controlled energy settings, the laser does not seem to produce implant surface alterations [21]. A recent study observed that the Er, Cr: YSGG laser may cause implant surface changes when the irradiation is set to 1.75 W/pulse or higher on the panel energy [19].

In more complicated peri-implantitis cases, complicated bony defects may exist, including supracrestal and intrabony defect components. For the management of supracrestal aspects, surgical elimination of pathological peri-implant pockets in combination with implantoplasty (IP) has been introduced [22,23]. Resective therapy with implantoplasty seems to positively impact the survival of oral implants affected by inflammatory processes [24]. However, adjunctive use of bone augmentation procedures and the principle of guided bone regeneration are suggested for intrabony components. During the surgical treatment, surface decontamination is thought to be an important step, and lasers are one of the effective surface decontamination methods. Microbiological and microscopy results indicate that the Er: YAG laser has high bacterial potential on common implant surfaces [13]. However, there have been no reports discussing the combination treatment of implantoplasty and Er, Cr: YSGG laser irradiation for the treatment of the implant surface.

Therefore, the purpose of this in vitro study was to establish a protocol for the suitable power output setting for an Er, Cr: YSGG laser based on the morphological changes and surface roughness of the virgin implant versus a surface that was already treated with implantoplasty. The null hypothesis was that Er, Cr: YSGG laser had an effect on the post-IP implant surface, including morphologic changes and compositional and surface roughness analysis.

## 2. Materials and Methods

### 2.1. Samples

Ten commercially available titanium dental implants (NobelReplace Conical Connection PMC, 4.3 mm diameter, 11.5 mm length, 0.71 mm thread pitch, TiUnite anodized implant surface, Nobel Biocare, Goteborg, Sweden) were used in the in vitro experiments. According to the manufacturer, the NobelReplace dental implants are made from biocompatible commercially pure grade 4 titanium and consist of a moderately rough thickened titanium oxide layer TiUnite surface. Five implants were used in experiment 1 (Non-IP group), and another five implants were used in experiment 2 (IP group).

### 2.2. Laser System

The laser system used was an Er, Cr: YSGG laser device (Waterlase, Biolase Technology, Inc., San Clemente, CA, USA), wavelength 2.78 μm, pulse repetition rate 20 Hz. For irradiation, completely new straight sapphire contact tips (600 μm in diameter, MGG6) were utilized with the above laser device. Energy settings on the panel were calibrated by the original dealer, measuring the output energy at the tip end using a power meter (Ophir power meter NOVA II, Ophir Photonics, Jerusalem, Israel).

### 2.3. Experiment 1: Morphologic Changes of the Microstructured Fixture Surface of Titanium Implants Following Irradiation (Non-IP Group)

To simulate clinical use, we performed sweeping movements on the microstructure surface of one virgin implant in noncontact mode at approximately 1 mm. The implant surface was circumferentially divided into four treatment areas on the long axis (Figure 1), and we performed the irradiations in the middle of each treatment area at a 2 mm distance from the neighboring laser area. A sweeping motion was performed by a well-trained dentist using an implant fixating tool, which was designed to ensure the same irradiated distance between the laser tip and implant surface (Figure 2). A moving length of 5 mm incorporated five repeated cycles for a total of 5 s. The different output energy values were 0.5, 1, 1.5, and 2 W on the panel, with a water spray at 20 Hz. Four implants were treated according to the above laser energy settings, and the same energy setting was repeated four times on each implant. Another virgin implant without laser irradiation was used for observation as a control. A total of five implants were used in experiment 1.

### 2.4. Experiment 2: Morphologic Changes of the Microstructure Fixture Surface of Titanium Implants Following Implantoplasty Plus Irradiation (IP Group)

We divided the implant surface into four equal treatment areas on the long axis (Figure 1). The structured surfaces were treated with the IP procedure, which included completely planishing and smoothening in the order of football-fine (grit size 46 μm) and extra fine (grit size 25 μm) grit diamond burs and round Arkansas stones (Shofu Inc., Kyoto, Japan) in a red contra-angle handpiece at about 120,000/6000 rpm under cooling with normal saline, according to Schwarz et al. [25]. An experienced dentist had practiced the same IP procedure on 20 implants for calibration before a formal test was executed, and all the pretest samples were carefully examined under 2.5× magnification dental surgical loupes with a headlight (PeriOptix Inc, Lompoc, CA, USA) to ensure that same quality of IP. All procedures were performed by the same experienced and trained dentist. The IP procedure was provided until the operator felt that the exposed threaded and structured areas were fully polished and smoothened, and the implant thread was not visible under 2.5× magnification dental surgical loupes with a headlight (PeriOptix Inc, Lompoc, CA, USA). After the IP procedures were completed, the laser was performed on each smooth surface on the long axis vertically, and sweeping movement was also employed using an implant fixture fixating tool, as in experiment 1. A vertical length of 5 mm was incorporated with five repeated cycles for a total of 5 s. Each was irradiated at 0.5, 1, 1.5, and 2 W set on the panel energy with a water spray at 20 Hz for 5 s. Four implants were treated according to the above laser energy settings, and the same energy setting was repeated four times on each implant. Another virgin implant which underwent only the implantoplasty protocol, without laser irradiation, was used as the control in experiment 2. In total, five implants were used in experiment 2. Figure 3 shows the outline of experiments 1 and 2.

### 2.5. Morphologic Analysis

In all experiments, the tested sites and surface alterations were assessed by optical stereomicroscopy (VHX-900F, Keyence Corp., Osaka, Japan) and scanning electron microscopy (JSM-6360, JEOL Ltd., Tokyo, Japan). In the stereomicroscopy, the prepared implant surfaces were inspected at a magnification of 200 times in experiments 1 and 2. In the SEM observations, the prepared implant surfaces were inspected at a magnification of 1500 times in experiments 1 and 2.

### 2.6. Compositional Analysis

Elemental analysis of the treated surfaces was performed using energy-dispersive X-ray spectroscopy (EDS, JSM-6360, JEOL Ltd., Tokyo, Japan). In experiments 1 and 2, carbon (C), oxygen (O), titanium (Ti), and phosphorus (P) contents on the treated surfaces were assessed. In experiment 1, one plain area (300 μm × 300 μm, Figure 4a) was randomly selected by the engineer from the affected surface between the second and third thread and subjected to area scan analysis. In experiment 2, one plain area was randomly selected by the engineer from each smooth, treated site on the same level and subjected to area scan analysis in an area of 300 μm × 300 μm (Figure 4b).

### 2.7. Surface Roughness Analysis

For the evaluation of the roughness parameters, a 600 μm × 600 μm plane area between the second and third thread from each surface was scanned on an optical profiler (Opt-scope R200, ACCRETECH Ltd., Tokyo, Japan). The files from the process were analyzed using dedicated software (Surfcom Map Opt ver.8.0.9216, Tokyo Seimitsu Co. Ltd, Tokyo, Japan). The implant specimen was mounted in a stainless steel mold, and then the profilometer tip was scanned along the middle of the treated implant surface. Each treated surface had two sites randomly selected by the software engineer and was recorded for mean values (Figure 4c). We set the implant target surface to the optical measuring instrument. The fixture and the instrument were not fixed. We manually moved the target area to be observed under the lens and aligned by eyes. During this process, deviations of a few micrometers to millimeters were created at each location. Thus, we completed the random selection. The following roughness parameter Ra was then calculated: 

Ra (arithmetic mean roughness) is the mean of the absolute values of the modified roughness profile, based on the central line to a reference route.

### 2.8. Statistical Analysis

To determine the required sample size, a post hoc power analysis was performed. The power calculation revealed that four treatment sites in each group (five groups in experiment 1 and five groups in experiment 2) reached a statistical power larger than 90% with a 0.05 level of significance.

In the EDS analysis of experiments 1 and 2, the mean values of the carbon, oxygen, phosphorus, and titanium contents were calculated. The Kruskal–Wallis test was used for statistical evaluation of the significance of differences between the three groups, and the Mann–Whitney U test was used for post hoc pairwise comparisons. The p-values were adjusted by using the Benjamini and Hochberg method.

In the surface roughness analysis of experiments 1 and 2, the mean values were calculated for the parameter of Ra. The Kruskal–Wallis test was used for statistical evaluation of the significance of differences between the three groups, and the Mann–Whitney U test was used for post hoc pairwise comparisons. The p-values were adjusted by using the Benjamini and Hochberg method.

## 3. Results

### 3.1. Morphologic Analysis of Stereomicroscopy and SEM

Figure 5 and Figure 6 reveal the image from the stereomicroscope and SEM in experiment 1. The picture of the nonirradiated control TiUnite surface is shown in Figure 5a. No obvious surface change occurred when it was irradiated at a panel energy of 0.5 W with a sweeping movement, compared with the untreated surface, according to the stereomicroscope and SEM (Figure 5b and Figure 6b). Nevertheless, the implant surface was affected when the panel energy was elevated to 1 W, 1.5 W, and 2 W. A partial melt of the microstructure was noted, as shown in Figure 5c and Figure 6c. The stereomicroscope also showed a blue –purple color change (Figure 5e) when 2 W irradiation was performed. SEM showed a melted surface and decrease in microporosity of the irradiation area at 1 W and 1.5 W (Figure 6c,d). Fusion of the pores and microcracks was observed at 2 W panel energy under SEM (Figure 6e). Overall, the severity of surface changes became more evident when the output energy was set to 1 W or higher.

Figure 7 and Figure 8 reveal the image from the stereomicroscope and SEM in experiment 2. All surfaces underwent implantoplasty before laser irradiation. A picture of the nonirradiated control of implantoplasty is shown in Figure 7a, revealing scratches and traces produced by polishing burs. The TiUnite porous architecture was fully removed by the implantoplasty procedure. Both stereomicroscope and SEM revealed no noticeable color or surface changes at any energy setting, including 0.5 W, 1 W, 1.5 W, and 2 W, compared with the nonirradiated control (Figure 7b,e and Figure 8b–e). It can be observed that the laser had literally no effect on the implant surface after the implantoplasty procedure, which resulted in the same scratching to the surface without any surface melting or microcracks.

### 3.2. Compositional Analysis of EDS

Overall, no other elements except carbon (C), oxygen (O), phosphorus (P), and titanium (Ti) on the fixture surface were detected. The element weight of C did not show a correlation with the panel energy in experiment 1 (Non-IP group), while O and P decreased, and Ti increased when the energy of the laser irradiation increased. When the panel energy was set to 0.5 W, no significant change was shown, except Ti showed a significant difference compared with the control. When the panel energy was set to 1 W or higher, significant changes in O, P, and Ti were shown compared with the control, while C did not show significant changes at any panel energy compared with the control in experiment 1 (Table 1).

No significant differences in elements C, O, P, or Ti at any panel energy setting were shown compared with the control in experiment 2 (Table 2).

### 3.3. Surface Roughness

In experiment 1 (Non-IP group), when the panel energy was set to 2 W, a significant increase in the Ra value was shown compared with the control (Table 3). A tendency towards an increasing Ra value when the panel energy increased was also noted (Figure 9).

In experiment 2 (IP group), each energy group showed a consistent performance of the Ra value (Table 4). No significant difference was shown at various energy settings. 

Overall, the implantoplasty procedure in experiment 2 revealed significantly lower Ra values when compared with all groups in experiment 1 at each laser irradiation energy setting. The lowest Ra value was obtained in the control of experiment 2.

## 4. Discussion

Over the years, many treatments for peri-implantitis have been proposed. Both laser and implantoplasty have been widely discussed treatment options for peri-implantitis. Waterlase laser, a class IV Er, Cr: YSGG laser with a 2780 nm wavelength (Biolase^®^, Irvine, CA, USA), is an ideal all-tissue laser because all dental tissues contain water. It delivers the highest level of clinician control, operating efficiency, flexibility in the tip, and accessory selection and is suitable for dental use, including caries removal, periodontal treatment, and other soft and hard tissue surgery, including surgical debridement of peri-implantitis. Scarano et al. [26] concluded that 1.5 W of Er, Cr: YSGG laser energy is recommended to be used in implant surface detoxification. Strever et al. [27] suggest that ≥95% of P. gingivalis were removed from SLA implant surfaces when the Er, Cr: YSGG laser power setting between 1 to 1.5 W was used. Eldeniz et al. [28] reported that 0.5 W of Er, Cr: YSGG laser reduced the viable microbial population in root canals. In the case of decontamination of the implant surface by laser irradiation, it is necessary to determine which parameters should be used so as not to affect the different implant surfaces.

On the other hand, implantoplasty is the mechanical modification of the implant, including thread removal and surface smoothening, and has been proposed during surgical peri-implantitis treatment, which was first mentioned by Rimondini et al. [29], and modified using different protocols by different authors [24,30,31]. Implantoplasty is an effective alternative to removing biofilm and reducing roughness. However, there is still no gold standard for treating peri-implantitis, and few articles have discussed the combined treatment of Er, Cr: YSGG laser and implantoplasty. It is a novel idea to combine these two different treatments since peri-implant defects are complicated, and more than one treatment may be used.

Under the observation of stereomicroscopy and SEM, no obvious change was noted at panel energy of 0.5 W with a sweeping movement and water irrigation, which means surface integrity was maintained when the panel energy was set to 0.5 W or lower. When the panel energy was elevated to 1 W or higher, the TiUnite implant surface was altered and showed a pronounced melted surface, which means the TiUnite surface cannot endure Er, Cr: YSGG laser at a 1 W or higher energy setting with a sweeping movement and water irrigation. The implant treated with implantoplasty revealed no obvious change at any energy level under stereomicroscopy and SEM, which means the core material of the implant body can tolerate Er, Cr: YSGG irradiation at energy settings from 0.5 W to 2 W under water irrigation. On the other hand, Takagi et al. [19] reported that slight color changes were observed by stereomicroscopy, and partially melted microstructures were found by SEM for sandblasted, acid-etched implants (SLA, Straumann) when 1.75 W/pulse energy was used. Taniguchi et al. [12] found that the 50 mJ/pulse Er: YAG laser without water produced moderate to severe changes on the implant surfaces, including Osseotite, Tioblast, SLA, RBM, and TiUnite. Chegeni et al. [21] pointed out that 1.5 W/pulse with 30 Hz does not seem to produce implant alterations which were sandblasted by alumina and oxidized with calcium and phosphorus. According to our in vitro tests, the TiUnite surface seemed to be less resistant to Er, Cr: YSGG laser compared with the above studies. It should be noted that the energy-setting threshold, which can cause the surface changes was different, and it may be speculated that a different surface treatment of the implant fixture was used, with an effect on resistance to the Er, Cr: YSGG laser irradiation.

Analysis of compositional changes indicated that microsurface changes before damage to the TiUnite surface could be observed. In experiment 1, the element mass concentration (%) of Ti increased when the panel energy was set to 0.5 W. No obvious change was observed under the stereomicroscope and SEM observation at 0.5 W; however, EDS analysis seems to be a more sensitive tool for detecting microsurface changes in TiUnite implants. More applications may be possible for intraoral implant surface changes when clinical implantitis occurs and is treated with Er, Cr: YSGG laser. The decrease in phosphorous when the panel energy was increased may have had an impact on the osteointegration of the TiUnite since phosphorus plays a role in bone conduction in the osteointegration reaction. In experiment 2, different energy values had no significant effect on the elemental composition of each group. Ti comprised most of the mass concentration since the TiUnite implant surface treatment was polished by the implantoplasty procedure, and thus, the core material of the Ti alloy was exposed for detection in the EDS analysis.

To determine the roughness of the implant surface treated with Er, Cr: YSGG irradiation and implantoplasty, the Ra values were recorded with an optical profiler. The TiUnite surface (Nobel Biocare, Gothenburg, Sweden) consists of a moderately rough thickened titanium oxide layer, with a roughness of 1.35 microns according to the official documentation. In experiment 1, the control group of implant surface roughness showed a similar roughness value as the official data. As the energy of laser irradiation increased, an increasing Ra was also noted, which could mean the laser damaged the surface integrity of the TiUnite surface and caused an irregularity. When the panel energy was set to 2 W, the Ra value showed a significant increase compared with the control in experiment 1, which also corresponded to the extended surface change observed by the stereomicroscope and SEM.

In experiment 2 of the IP group, each energy group showed a consistent Ra value and no significant difference between various laser energy settings, which means laser irradiation did not have an impact on the surface roughness of the post-implantoplasty surface. It might be speculated that the implantoplasty surface is smooth enough to resist Er, Cr: YSGG irradiation. Studies have shown that below the Ra value of 0.2 microns with a machined surface, no influence on the quantity and composition of the biofilm is proven [32,33]. The Ra value in the present study ranged from 0.621 to 0.771 microns, which is higher than the 0.2 microns threshold for Ra, suggesting the present implantoplasty protocol did not achieve the level of smoothness of the moderately roughened TiUnite surface. Additional surface treatment with polishers is required to achieve the desired Ra value of 0.2 microns [34,35]. Although the Ra value of the IP group was lower than that of the non-IP group for all laser energy settings, the implantoplasty procedure alone was not adequate for the prevention of the re-establishment of oral biofilm. Other implantoplasty protocols are also mentioned for decreasing surface roughness, but an optimal balance between final surface roughness and overall treatment time should be noticed [30]. Implantoplasty, although technically changing the surface composition of a dental implant, does not seem to significantly alter the fracture resistance of standard-diameter external-connection implants [36]. Implantoplasty seems not to be associated with any remarkable mechanical or biological complications in the short to medium term [37], and such a procedure applied to implants has a chemical elemental composition comparable to the machined implant surface, according to Schwarz et al. [25]. However, implantoplasty of diameter-reduced implants should be more cautious since it decreases implant wall thickness and fracture resistance and varies depending on the implant–abutment connection [38].

The outcomes showed no significant difference in compositional and surface roughness analysis when Er, Cr: YSGG laser was applied to a post-IP TiUnite surface implant. The null hypothesis was rejected.

The limitation of the present study was that the samples were all virgin implants, and the laser irradiation was performed in a lab, which may not coincide with the oral cavity. Results are only indicated on the TiUnite anodized surface implant.

## 5. Conclusions

From the perspective of maintaining the integrity of the TiUnite surface coating, the Er, Cr: YSGG laser should be set to 0.5 W or below on the panel energy with water spray at 20 Hz since 1 W or higher may cause TiUnite surface damage while 0.5 W~2 W laser energy did not have an effect on the post-implantoplasty surface. EDS analysis can be used as a predictor for surface element composition change under laser irradiation and implantoplasty, showing a compositional change when the Er, Cr: YSGG laser panel energy was set to 0.5 W or higher, whereas no significant difference in the element composition was found once the surface was polished with IP. The Er, Cr: YSGG laser made the TiUnite surface rougher at 2 W panel energy but did not alter the post-implantoplasty surface roughness. Implantoplasty plus laser irradiation is a novel treatment combination, but further research is recommended to test the efficacy of the present treatment protocol in clinical practice since there is still no gold standard for the treatment of peri-implantitis to date.

## Figures and Tables

**Figure 1 jfb-13-00133-f001:**
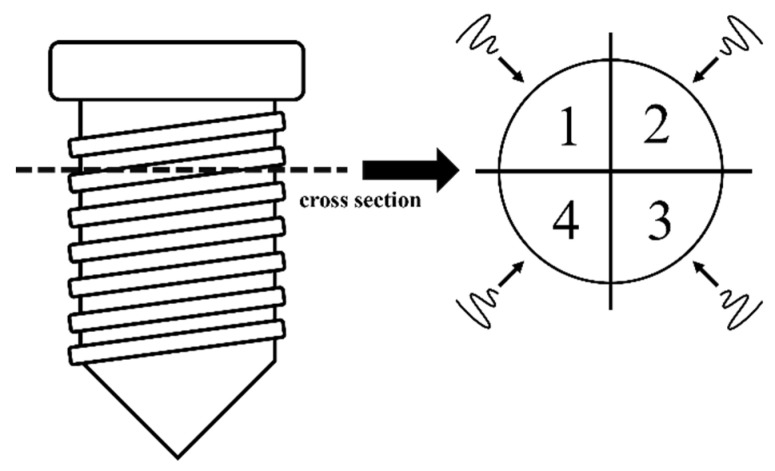
We divided the implant surface into four areas on the long axis, and laser irradiations were performed on each site.

**Figure 2 jfb-13-00133-f002:**
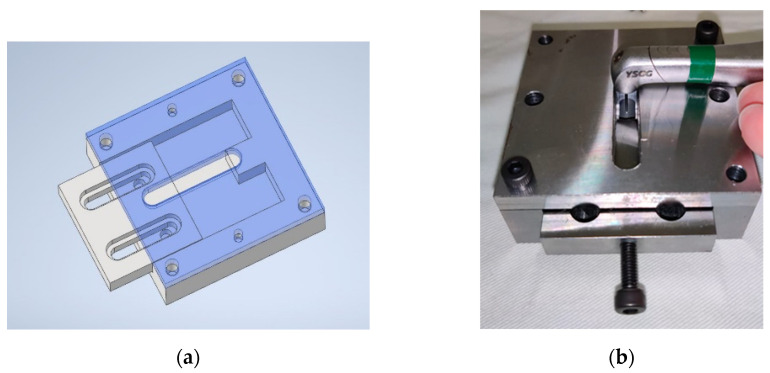
A sweeping movement was performed using an implant fixating tool, which was designed to ensure the same irradiated distance between the laser tip and implant surface: (**a**) CAD design of the tool; (**b**) the picture of performing laser irradiation using the tool.

**Figure 3 jfb-13-00133-f003:**
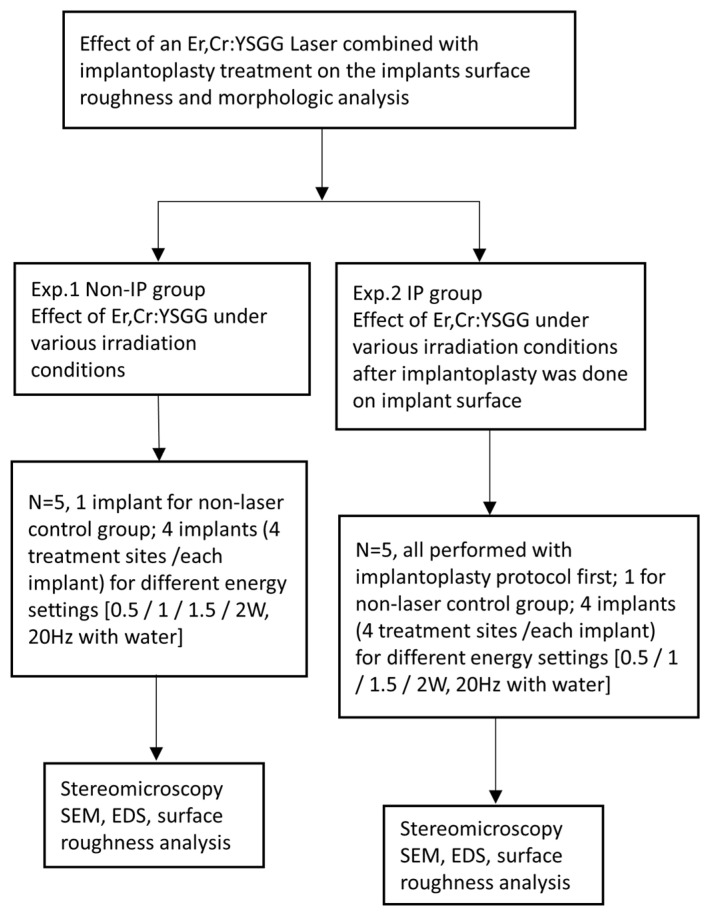
Outline of experiments 1 and 2.

**Figure 4 jfb-13-00133-f004:**
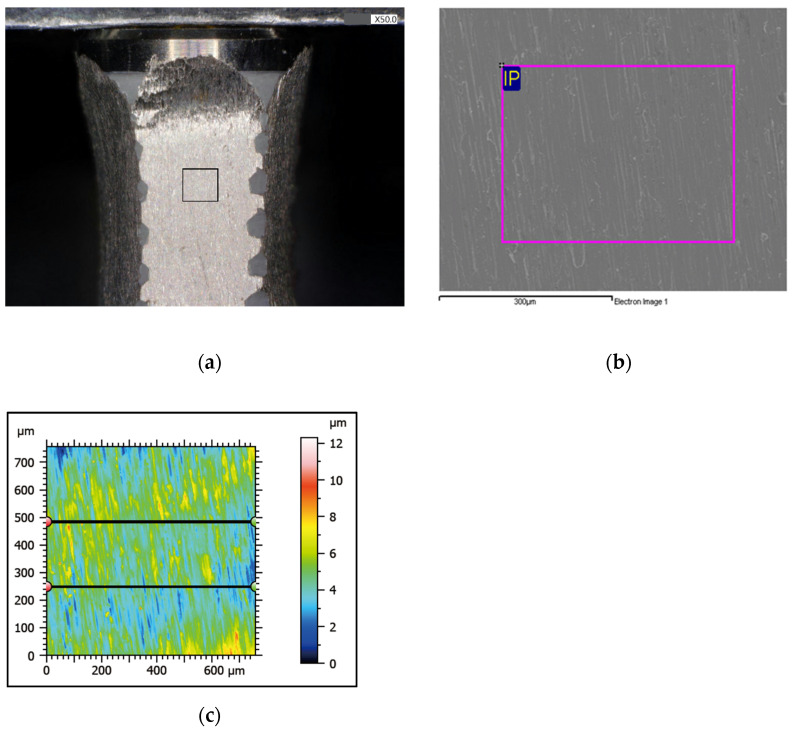
(**a**) Fifty times magnification under the stereomicroscope, one plain area (300 μm × 300 μm) was randomly selected from the affected surface between the second and third thread and subjected to area scan analysis.; (**b**) EDS analysis of a chosen area between the second and third thread; (**c**) surface roughness evaluation: each treated surface had two sites recorded for mean values.

**Figure 5 jfb-13-00133-f005:**
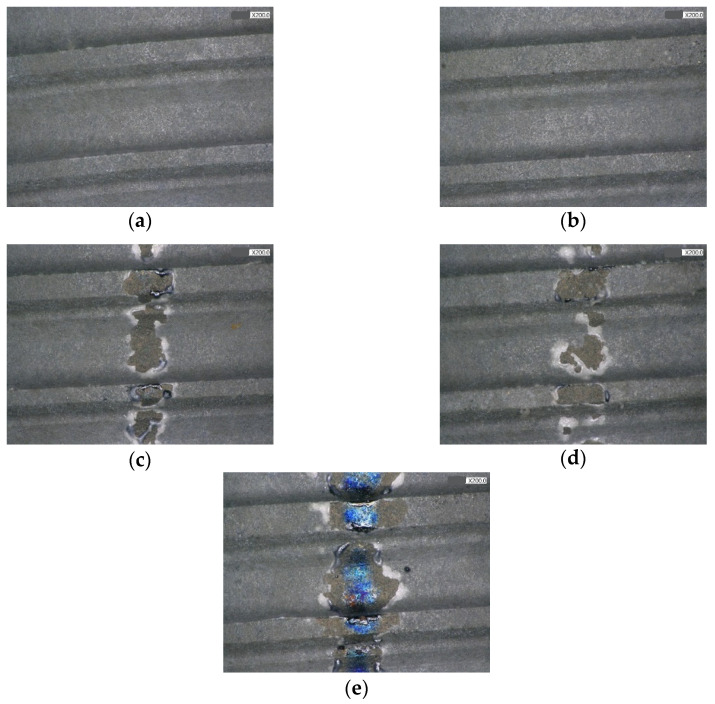
Non-IP group observation under stereomicroscope, 200×; (**a**) control group; (**b**) 0.5 W; (**c**) 1 W; (**d**) 1.5 W; (**e**) 2 W.

**Figure 6 jfb-13-00133-f006:**
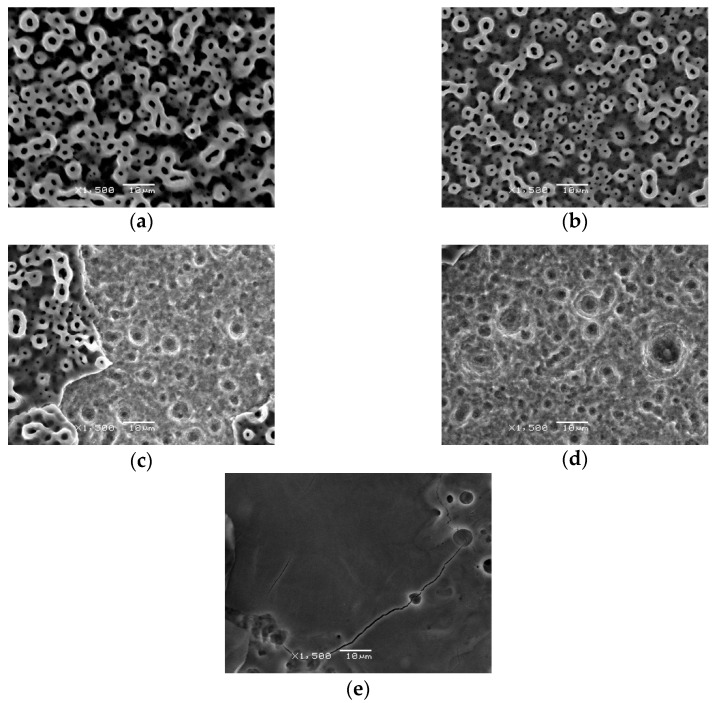
Observation under SEM, 1500×; (**a**) control group; (**b**) 0.5 W; (**c**) 1 W; (**d**) 1.5 W; (**e**) 2 W.

**Figure 7 jfb-13-00133-f007:**
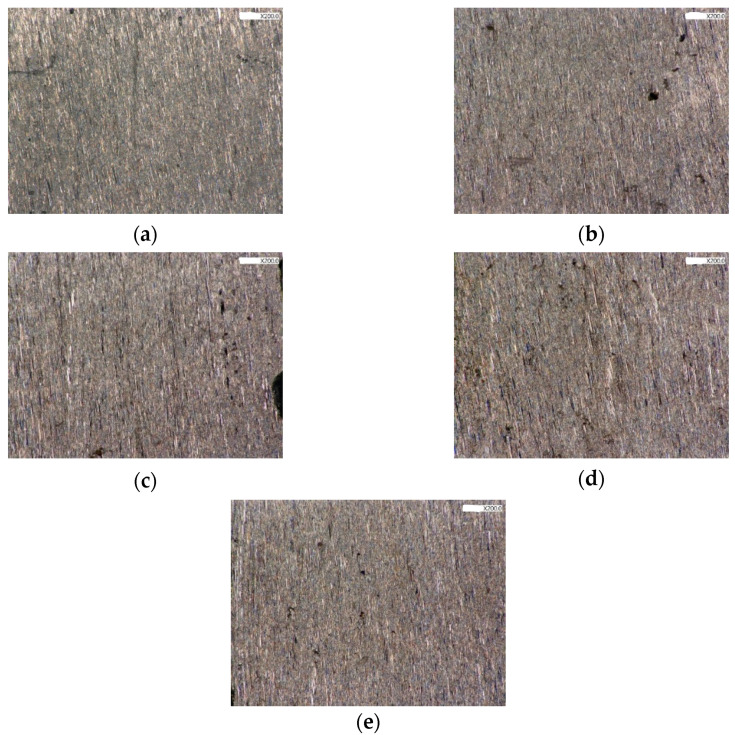
Observation under stereomicroscope, 200×; (**a**) IP control group; (**b**) IP + 0.5 W; (**c**) IP + 1 W; (**d**) IP + 1.5 W; (e) IP + 2 W.

**Figure 8 jfb-13-00133-f008:**
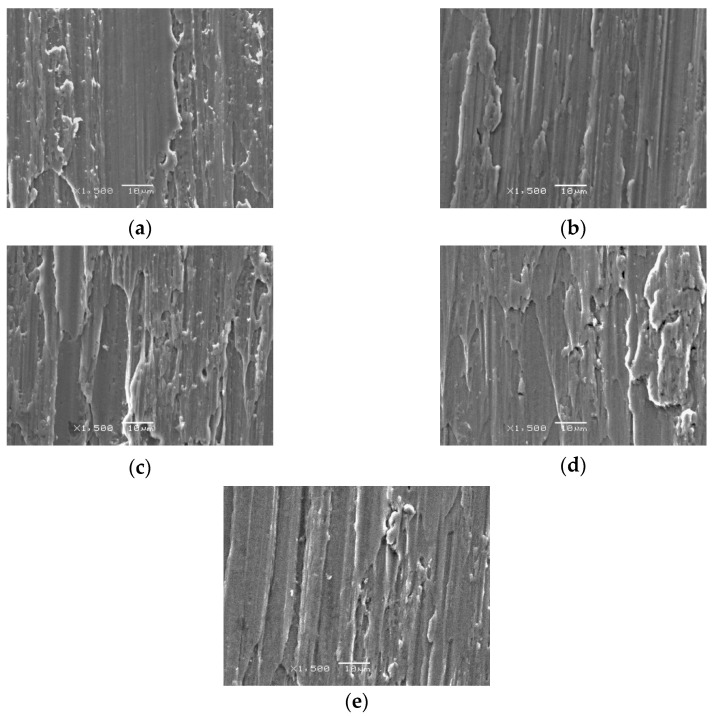
Observation under SEM, 1500×; (**a**) IP control group; (**b**) IP + 0.5 W; (**c**) IP + 1 W; (**d**) IP + 1.5 W; (**e**) IP + 2 W.

**Figure 9 jfb-13-00133-f009:**
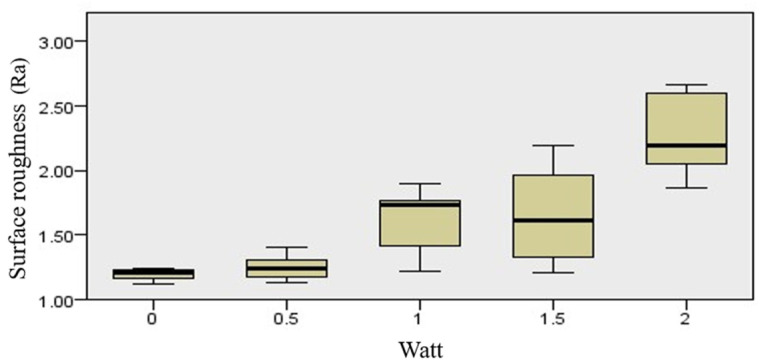
Tendency towards increasing Ra value when the panel energy was increased, *p* < 0.05.

**Table 1 jfb-13-00133-t001:** Element weight (%) of Non-IP group in experiment 1.

	C	O	P	Ti
Control	2.94	35.32	7.82	53.92
0.5 W	2.40	33.21	7.15	57.25 *
1 W	2.96	29.49 *	5.63 *	61.93 *
1.5 W	2.54	30.05 *	5.37 *	62.04 *
2 W	3.05	22.74 *	2.13 *	72.08 *

* Statistically significant difference between test group and control group, *p* < 0.05.

**Table 2 jfb-13-00133-t002:** Element weight (%) of IP group in experiment 2.

	C	O	P	Ti
IP Control	3.63	3.63	0.10	92.64
IP + 0.5 W	3.39	1.09	0.00	95.53
IP + 1 W	2.87	1.61	0.00	95.52
IP + 1.5 W	2.66	3.42	0.00	93.92
IP + 2 W	2.34	2.62	0.00	95.04

**Table 3 jfb-13-00133-t003:** Surface roughness of Non-IP group in experiment 1.

	Ra Value (Mean ± SD)	95% CI	
Control	1.194 ± 0.06	1.165	1.165
0.5 W	1.244 ± 0.091	1.181	1.181
1 W	1.617 ± 0.237	1.453	1.453
1.5 W	1.652 ± 0.383	1.386	1.386
2 W	2.277 ± 0.309 *	2.063	2.063

* Statistically significant difference between the test group and control group, *p* < 0.05.

**Table 4 jfb-13-00133-t004:** Surface roughness of IP group in experiment 2.

	Ra Value (Mean ± SD)	95% CI	
IP Control	0.621 ± 0.06	0.579	0.663
IP + 0.5 W	0.709 ± 0.091	0.646	0.772
IP + 1 W	0.674 ± 0.157	0.565	0.783
IP + 1.5 W	0.728 ± 0.267	0.544	0.913
IP + 2 W	0.771 ± 0.134	0.679	0.864

## Data Availability

Correspondence and requests for materials should be addressed to Aaron Yu-Jen Wu.

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
