# Peer review of "The Effect of an Er, Cr: YSGG Laser Combined with Implantoplasty Treatment on Implant Surface Roughness and Morphologic Analysis: A Pilot In Vitro Study"

_jfb, 2022, doi:10.3390/jfb13030133_

Round 1
Reviewer 1 Report
This is a pilot in vitro study and should be specified in the title, since no sample size calculation is provided based on data from previous studies or the difference in effect size that can be considered clinically relevant between the two. tested groups.
On the other hand, the study is justified since, in the clinical scenario, access to the implant surfaces to perform a correct implantoplasty is complicated due to the morphology of the defects and, in this case the complementary use of laser can be justified.
Introduction
It must include a definition of peri-implantitis according to the consensus of the 2017 workshop.
The statement in lines 39-40 is not true, since before an accumulation of plaque and calculus occurs on the surface of the implant, there must be a loss of bone height that not always corresponds to implant pathology. The paragraph should be improved.
Material and methods
The characteristics of the body and the surface of the tested implant (commercially pure Ti and grade and the characteristics of the surface) should be better described in this section. It should be considered that the results obtained in the study are only valid for this type of implant and surface.
Authors must explain how the researcher is calibrated in both experiments 1 and 2, it is really difficult to obtain a uniform result of all the samples regarding the implantoplasty since it is a "perception" of the operator.
A single implant as a control with and without implantoplasty may not be adequate.
It should be explained how randomize the analyzed area.
For the surface roughness analysis, must be described whether the area was randomly selected.
In line 162, a description of the commercial name and manufacturer of the software used for the analysis of the profile should be added.In figure 4 it can be seen that the side of the implant treated with implantoplasty (left) has not undergone a total removal of the implant threads as described in the Schwartz protocol.
Results
They are well reported.
Discussion
The authors should argue whether 0.5 W is enough to achieve bacterial decontamination.
Authors should reason more extensively about the additional advantages of the combined IP+Laser protocol, since this statement cannot be made if bacterial cultures are not performed, in order to check if there is a greater bacterial decontamination, a delay in recontamination or a change in the composition of the biofilm.
Row 261: specify with bibliographic references of the different authors who propose the implantoplasty protocols and contrast the degree of polishing with that obtained in this experimental in vitro study.
Row 323: must add that, although it is true that it does not greatly alter the resistance to fracture after an implantoplasty in regular diameter implants, this does occur in narrow diameters and especially in cases of internal connection (the references must be added corresponding bibliographic references (Camps-Font et al.)
It would be interesting to add data on the effects of laser on other implant surfaces and compare with the results obtained in this study.
A section on limitations of the study is needed
Conclusions
Are correct and correspond to the results obtained
Author Response
Thank you for your comments.
The changes in the manuscript uploaded by reviewers have been accepted and marked in red font. Please see the attachment.

Reviewer 2 Report
TITLE: The effect of an Er, Cr: YSGG Laser combined with implantoplasty treatment on implant surface roughness and morphologic analysis.
The aim of the present investigation was to evaluate a protocol for the suitable power output setting for an Er, Cr: YSGG laser, based on the morphological changes and surface roughness of the virgin implant versus a surface which was already treated with implantoplasty.
GENERAL COMMENTS
The article is in-line with the journal topic, but flaws should be improved. The investigation is interesting, and the present paper is recommended for publication to the present journal after major revision.
Title: The title should indicate the type of study that has been conducted. (in vitro, in vivo)
Introduction
1. The effect of erbium-doped yttrium aluminum garnet laser (er: yag) irradiation on sandblasted and acid-etched should be discussed in this section (PMID: 32961798; PMID: 32962189).
2. The peri-implantitis and mucositis etiology, risk factor and stadiation should be introduced.
3. The null hypothesis is missed..
Materials and methods
1. The main characteristics of the implants used for this study should be described (surface treatments, micro and macro geometry, thread pitch and profile…).
2. Did you performed a sample size calculation to determine the number of specimens?
3. The statistical methods should be indicated in a separated sub-paragraph and expanded.
Results
The results presentation is a little bit poor. Why did you choose the same magnification for both SEM and stereomicroscope? The authors should add SEM images with a higher magnification in order to examinate in detail the surface alteration determined by the laser irradiation (microcracks…).
The Ra values descriptive statistics is missed in this section (mean, SD, 95% CI…). The p value is missed in fig.9.
Discussion
The present investigation produced a clinical evaluation with no randomization/ blinding protocols with a high risk of bias. This important aspect should be discussed in this section. In my opinion, an improved analytical approach could strongly potentiate the study findings.
The null-hypothesis should be discussed in this part of the manuscript.
References
A high quantity of references is outdated and a more recent bibliography is strongly recommended.
Author Response

(The authors gave the same response as above.)

Round 2
Reviewer 1 Report
Introduction
Must replace reference 3 of the bibliography with the most up-to-date:
Peri-implant diseases and conditions: Consensus report of workgroup 4 of the 2017 World Workshop on the Classification of Periodontal and Peri-Implant Diseases and Conditions
https://doi.org/10.1111/jcpe.12957
Methods
Regarding the description of the randomization of the analyzed surfaces, it is not only important that they describe who did it, but also how they did it, that is, the randomization method.
Bibliography
All bibliographic citations must be reviewed completely. Vancouver-style has very clear rules and the authors can find different tutorials on the net.
- The first 6 authors must be cited and only if there are more, et al.
- Nor can some authors be written with the surname and the name initials and others, only with the initials of both the name and the surname.
- The p of pages is left over.
- After the name of the authors should not put a comma but a point.
Author Response
Thanks very much for taking your time to review this manuscript. I really appreciate all your comments and suggestions! The responses were marked with red font. Please see the attachment.

Reviewer 2 Report
the paper has been improved
Author Response
Thank you for that excellent and insightful series of remarks.